# Modelling the health labour market outlook in Kenya: Supply, needs and investment requirements for health workers, 2021–2035

James Avoka Asamani[1,2]*, Brendan Kwesiga[3], Sunny C. Okoroafor[1], Evalyne Chagina[4], Joel Gondi[5], Zeinab Gura[5], Francis Motiri[5], Nakato Jumba[5], Teresa Ogumbo[5], Nkatha Mutungi[5], Stephen Muleshe[5], Yusuf Suraw[5], Hanah Gitungo[5], Kiogora Gatimbu[5], Mutile Wanyee[5], Amos Oyoko[5], Angela Nyakundi[5], Stephen Kaboro[5], Mary Wanjiru Njogu[5], Maureen Monyoncho[5], Njoroge Nyoike[6], Wesley Ogera Ooga[5], Juliet Nabyonga-Orem[2,7], Julius Korir[8], Paul Marsden[9], Mona Almudhwahi Ahmed[4], Julius Ogato[5], Pascal Zurn[9], Annah Wamae[5]

1 Health Workforce Unit, Universal Health Coverage—Life Course, World Health Organization Regional Office for Africa, Brazzaville, Congo, 2 Centre for Health Professions Education, Faculty of Health Sciences, North-West University, Potchefstroom, South Africa, 3 UNICEF Program Group-Health, Nairobi, Kenya, 4 World Health Organization, Kenya Country Office, Nairobi, Kenya, 5 Ministry of Health, Nairobi, Kenya, 6 Kenya Bureau of Statistics, Nairobi, Kenya, 7 World Health Organization, Namibia Country Office, Windhoek, Namibia, 8 University of Nairobi, Nairobi, Kenya, 9 Health Workforce Department, World Health Organization Headquarters, Geneva, Switzerland

* asamanij@who.int

## Abstract

Kenya is committed to achieving Universal Health Coverage (UHC) within its devolved health system in which significant investments have been made in health infrastructure, workforce development, and service delivery. Despite these efforts, the country faces considerable health workforce challenges. To address these, the Ministry of Health undertook a comprehensive Health Labour Market Analysis (HLMA) in 2022 to generate evidence supporting the development of responsive health workforce policies. This paper presents findings of a modelling exercise to understand the health labour market outlook. As part of a comprehensive HLMA, a validated needs-based health workforce modelling framework was applied to project the supply, needs, and investment requirements. Data was triangulated from multiple sources through desk reviews and group modelling by an expert technical working group constituted to undertake the study. The analysis considered disease burden, population growth, service delivery models, and health worker productivity, to assess the future health workforce needed. Kenya's health workforce is growing, with approximately 7,650 new workers added annually, resulting in an estimated 3.4% annual growth. By 2025, the health workforce is projected to reach 226,434, increasing to 263,700 by 2030. However, Kenya required a minimum of 254,220 health workers in 2021 to make substantial progress toward UHC. The cumulative need could rise to 476,278 by 2035. In 2021, Kenya had a needs-based shortage of nearly 60,000 health professionals, which could increase to 114,352 by 2030. The financial space for health workforce was estimated at US$2.29 billion in 2021 and is projected to rise to US$3.58 billion by 2030, but the required wage bill potentially reaching US$3.9 billion. Kenya must significantly

**Data availability statement:** All relevant data are publicly available within the paper and its Supporting Information files. The full health labour market analysis report is publicly available from the Republic of Kenya labour market observatory (https://labourmarket.go.ke/media/resources/Final_Kenya_HLMA_Report_2023_v8.pdf).

**Funding:** Data collection for this study was funded by the World Health Organization through ILO-OECD-WHO Working for Health (W4H) grant number 76677 and grant number 71753 from the Department of Foreign Affairs, Trade and Development (DFATD), Canada. JAA, SCO, BK and JNO were supported by both grants. The funders had no role in study design, data collection and analysis, decision to publish, or preparation of the manuscript.

**Competing interests:** The authors have declared that no competing interests exist.

increase investments in its health workforce to meet UHC goals. Both public and private sectors need to contribute more, with the public sector requiring a health workforce budget increase of 10.5% annually to bridge the projected funding gap.

## Introduction

Over the years, the WHO African Region has made modest strides in expanding its training and production capacity and its stock of health professionals. There are more than 4,000 institutions that provide health professions education and training. From 2005 to 2020, medical schools increased from 168 to 401, and there are at least 2,122 nursing and midwifery schools in 43 Member States where data is available. With the expansion of the training infrastructure, training output has increased proportionally. For instance, the number of trained physicians has increased from 6,000 per year in 2005 to more than 30,000 by 2022, and the number of trained nurses and midwives has increased from 26,000 per year in 2005 to at least 151,000 per year [1–3]. The WHO African Region's health workforce stock has risen to 5.1 million in 2022 compared to 4.3 million in 2018 [3].

Despite the progress in increasing the stock of health workers from 1.6 million in 2013 to 5.1 million in 2022, the WHO African Region needs more than 6.1 million alongside addressing the maldistribution of the available health workers to achieve UHC by 2030 [4,5]. As a result, 37 out of 55 countries on the WHO's Safeguard and Support List (SSL) are in the African region [6,7]. In addition, even though over 255,000 health workers are trained annually in Africa, absorption capacity (i.e., financial ability and willingness of countries to pay for health workers in its efforts to meet the population's health need) remains challenged due to limited fiscal space for health employment [3] and budgetary space analysis suggesting a 43% funding gap in the public sector to employ the current supply of health workers in East and Southern Africa [8]. With this, the health worker shortage in Africa is decreasing much more slowly than in the rest of the world [9], leaving nearly one-third of trained healthcare professionals in Africa facing lengthy periods of unemployment, underemployment, and transient or casual employment (termed precarious work). On average, 22% of health professionals in the WHO African region are employed in the private sector, but the private sector represents 40% of the training capacity [10]. Thus, the private sector still needs to reach its maximum employment potential within the health workforce.

Kenya has been implementing the Universal Health Coverage (UHC) programme as a major political priority for the entire country in the context of a devolved health system. In the previous Government (2018 to 2022), the main tenets of the UHC programme were to progressively increase the percentage of Kenyans covered with essential health services under prepaid health financing mechanisms such as health insurance, subsidies and direct government funding to access health services, expanding the scope of the health benefit package, improving the quality of health services, retaining health resources appropriate for the delivery of health services and strengthening the leadership and governance within the health sector [11–13]. Continuing with the same agenda of UHC, the new Government that came to power in 2022 emphasised the importance of Human Resources for Health, Primary Health Care, Health Commodity Security and Integrated Health Information System.

In line with the provisions of the Kenyan Constitution, which guarantees the right to health, the Government's Vision 2030, operationalised through the 5-year Medium-Term Plans, recognises health as one of the components of delivering its Social Pillar by maintaining a healthy and skilled workforce necessary to drive the economy. It prioritises the development and employment of the health workforce as one of the main projects to achieve its goals.

Over the last decades, Kenya has more than doubled the density of doctors, nurses, midwives and clinical officers per 10,000 population from 14.47 in 2006 to 30.14 in 2021. At an annual average growth rate of 7%, access to health workers is increasing, but unemployment is also rising among the skilled health workforce in Kenya– estimated at 14% in 2021 [14]. Meanwhile, there are growing concerns about an escalating health sector wage bill, which increased by 44% between 2014/2015 and 2018/2019, from 4.97 billion Kenyan Shillings in the 2014/2015 fiscal year to 7.14 billion Kenyan Shillings in the 2018/2019 fiscal year [15]. Even though unfilled vacancies are apparent within Counties, the drive to commit more resources towards health workforce job creation appears to be dwindling amidst concerns that health service outputs and outcomes have not been meeting expectations. The Government of Kenya, since October 2019, has been holding a national wage bill and sustainability conference each year [16] to explore approaches to achieve a target of a 35% wage bill-to-revenue ratio [17] without necessarily compromising service delivery and job creation for the youth.

Against this backdrop, the National Human Resource for Health (HRH) Strategic plan prioritised the need to conduct a health labour market analysis and set up a productivity management system to make an investment and efficiency case to foster dialogue towards job creation in the health sector. This study was part of a more extensive assessment of the health labour market in Kenya and sought to model the supply, need and budget space requirements to address potential health labour market mismatches. The descriptive aspects of the health labour market analysis have been discussed elsewhere [14].

## Methodology

### Modelling the future supply and needs-based requirements for health workers

The policy issues identified in Kenya for modelling the health labour market included determining (1) how many health workers are anticipated to be in the health labour market in the future, (2) how many health workers will be needed to address the health needs of the population, and (3) the absorption capacity of the health system for sustainable employment of health workers.

In undertaking the modelling, an empirical framework for integrated analysis of HWF supply, needs and economic feasibility was applied (Fig 1) [18] and leveraged a published simulation tool built in Microsoft Excel [19]. As health workforce modelling is complex and requires multi-dimensional skills, the Technical Working Group (TWG) worked with experts from various aspects of healthcare and other sectors, including epidemiologists, public health experts, clinicians from diverse backgrounds (doctors, nurses, midwives, clinical officers, pharmacists, laboratory scientist, and nutritionist among others), statisticians, economists, and human resource management. A group modelling approach was adopted in which the team worked in smaller teams in a dedicated two-week working session, with guidance and technical support from WHO technical experts.

Three specific estimations were made for the (a) supply of health workforce, (b) the needs-based requirements for health workforce and (c) cost and economic feasibility. The technical methodology for these estimations has been adequately described elsewhere in the literature [5,18–23], hence are briefly highlighted in this section.

**Modelling the supply of health workers.** Building on the stock and labour flow information on the health workforce, the future supply of health workers was projected using a stock-and-flow methodology, as illustrated in equation (1).

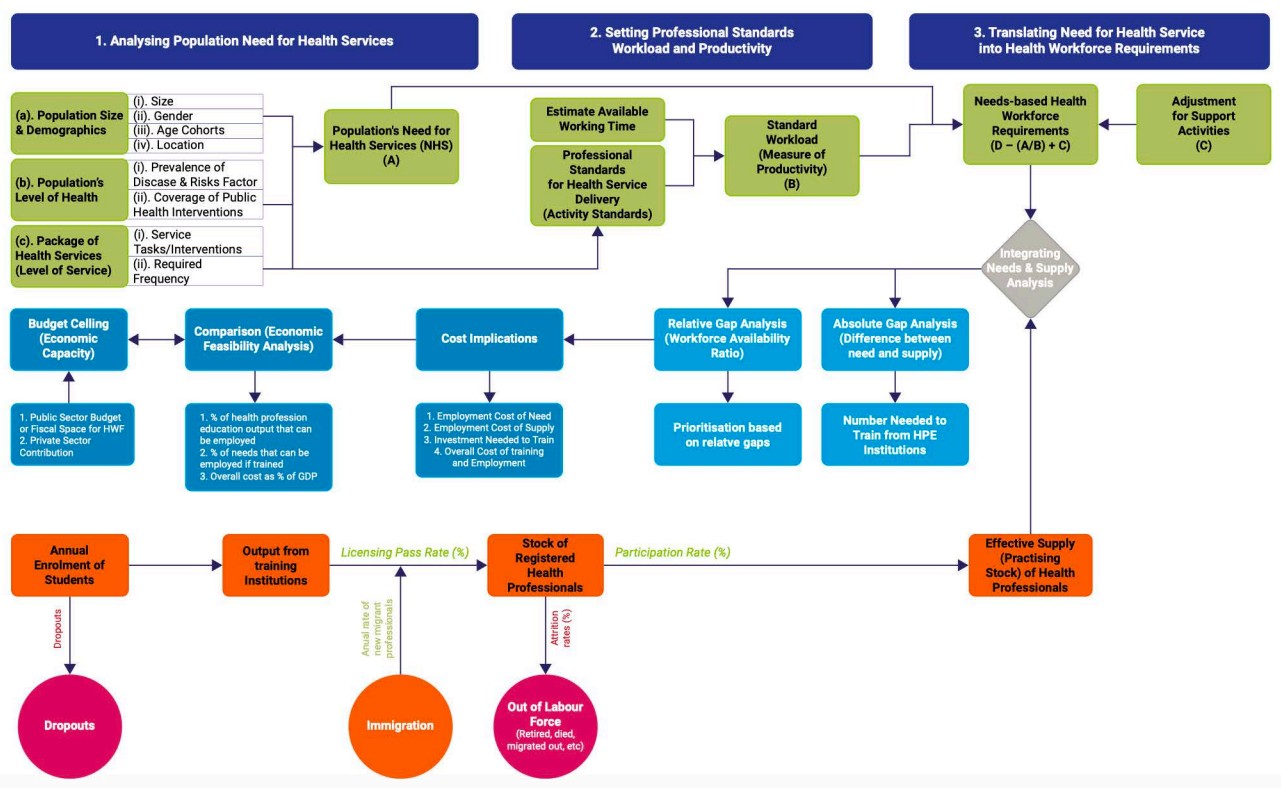

**Fig 1. Analytical framework for the supply and needs-based health workforce forecasting.** Source: adapted from Asamani et al. [19].

$$S_{n,t} = \left[ T_{n,t-1} \times \left(1 - a_n\right) + I_n \right] \times P \qquad \text{equation 1}$$

In this equation, $S_{n,t}$ refers to the supply of health worker of category n, at time t, $T_{n,t}$ is the aggregate stock of health worker of category n at time t, $a_n$ represents the attrition rate (a proportion of the stock, $T_{n,t-1}$ that died, retired, could not work due to ill-health or migrated out), $I_n$ is the inflows of health worker of category *n* trained domestically or immigrating from another country while *P* is the labour participation rate or the proportion of the health workers willing to engage in professional practice.

**Modelling the needs-based requirements for health workers.** There are several methods for determining the 'needed' health workforce in a country [24], but the Global Strategy on Human Resources for Health recommends a needs-based approach that aligns investments to population health needs [25]. The health needs-based or epidemiology approach was adopted with the assumption that the need for health workers in Kenya depended on the 'need for health services' as defined by the disease burden and structure of the population alongside the health service delivery model of the country [26,27]. The following technical steps were followed to determine the need for health workers.

*Estimating the populations' 'need for health services'.* It was prioritised to quantify the 'need for health service' that covers at least 95% of the burden of diseases and risk factors. The list of diseases and risk factors that account for 95% of morbidity and mortalities was identified using data from the country's health information system. A team of epidemiologists and statisticians then conducted a desk review to obtain the prevalence rates of the diseases

and risk factors and the targets of the coverage rates of priority public health interventions (S1 Data). The disease burden and the risk factors were mapped using the routine health information from the Kenya Health Information System (KHIS), Kenya Demographic Health Survey (KDHS), Kenya Household Survey, and Mid Term Review of the Kenya Health Sector Strategic Plan. Desk review of prevalence estimates was undertaken from the aforementioned sources and complemented with evidence peer-reviewed scientific papers. A separate team of clinical experts were divided into three groups – Communicable diseases (CD) team, Non-Communicable Disease (NCD) team, and Reproductive, Maternal, Newborn, Child and Adolescent Health (RMNCAH) team who worked together to identify the planned or other-wise necessary health intervention to address each of the diseases and risk factors identified as well as the health worker occupational group that has the competency to deliver the interventions. The team also identified the appropriate population cohorts (demographic groups, gender, and location) to benefit from the interventions (services). These clinician-led works were combined to compute the needs-based service requirements using equation 2.

$$NHS_t = \sum P_{i,j,g,t} \times [H_{h,i,j,t-1} \times (1+R_h)] \times L_{y,h,i,j,t} \qquad \text{equation 2}$$

In equation 2, **NHS**$_t$ represents the 'Needed Health Services' by a given population under a given service delivery model, $L_{i,j,t}$ over a period of time $t$, $P_{i,j,g,t}$ represents the size of the given population of age cohort $i$, gender $j$ in location (rural or urban) $g$ at time $t$ (i.e., population and its demographic characteristics), $H_{h,i,j,g,t}$ represents the proportion of the given population with health status $h$, of age cohort $i$, gender $j$ in location $g$ at time $t$ (i.e., the level of health of the population), $R_h$ is the instantaneous rate of change of the health status, $h$ while $L_{y,h,i,j,g,t}$ represents the frequency of health services of type $y$ planned or otherwise required, under a specified service model, to address the needs of individuals of health status $h$ among age cohort $i$, gender $j$ in location $g$ over time $t$ (i.e., the level of service required by the population).

***Translating the need for health service into needs-based staffing requirements***: Leveraging past and ongoing WISN studies in Kenya and augmented with experience from other countries, a standard workload was determined for each health intervention identified by the clinical expert teams (see equation 3). A standard workload, akin to a measure of productivity, is the volume of work within one health service activity that one health worker can accomplish within a year to acceptable professional standards [28]. The estimated "need for health services' was then translated into the health workforce using the standard work (see equation 4).

$$SW_{n,y} = \frac{AWT_n}{SS_{y,n}} \qquad \text{equation 3}$$

In equation 3, $SW_{n,y}$ represents the standard workload for health professionals of category $n$ when performing health service activity $y$ while $AWT_n$ represents the annual available working time of the health professional of category $n$ and $SS_{y,n}$ represents the Service Standard or the time it takes a well-trained health professional of category $n$ to deliver the service activity, $y$.

$$Needs-based\ HWF\ requirements_{n,y} = \sum \frac{NHS_{n,y,t}}{SW_{n,y}} \qquad \text{equation 4}$$

In equation 4, NHS$_t$ represents the number of needed health service activity $y$, to be delivered by a health professional of category $n$ at time $t$ while $SW_{n,y}$ is the standard workload for health professionals of category $n$ when performing health service activity $y$.

**Modelling the absorption capacity (financial space) for the health workforce.** The absorption capacity to employ health workers is reflected in a country's ability and

willingness to pay for health workers in its efforts to meet the population's health needs. Thus, the absorption capacity for health workers is a part of the joint financial capacity of the Government, development partners and the private sector in purchasing health care services, in which the cost of health workers' wages represents a substantial proportion [29]. The assumption underlying this approach is that countries (governments and partners) will not necessarily spend on healthcare more than they can afford, even if their health or level of health utilisation is suboptimal relative to internationally established metrics [30]. Therefore, demand for health workers can be gauged using fiscal space for the wage bill as a proxy and adjusting for the private sector contribution to HWF employment (equation 6). Analysis of the health sector budget was undertaken to gauge the level of prioritisation of the HWF within the successive budgets.

$$\begin{aligned} \textbf{Public sector HRH budget} \\ \textbf{space for the year, i} \end{aligned} = \begin{pmatrix} \text{GGHE as \% GDP}_i * \text{Nominal GDP Values}_i \\ \text{X HRH Expenditure as \% GGHE}_i \end{pmatrix} \qquad \text{equation 5}$$

$$\begin{aligned} \textbf{Cumulative financial space} \\ \textbf{for the year, i} \end{aligned} = \begin{aligned} \text{Public Sector Fiscal Space}_i \text{ X } (1 + \text{proportion of} \\ \text{private sector HRH employment}) \end{aligned} \qquad \text{equation 6}$$

Whereby *i* is the target year, **GGHE** represents the General Government Health Expenditure, and **GDP** is the Gross Domestic Product. In this equation, it was conservatively assumed that if the Government continued to spend a similar proportion of GDP on health and a similar proportion of GGHE on HRH, all other things being equal, the fiscal space for HRH would be proportional to the size of the GDP. It was further assumed that the private sector would not contract and that a conservatively similar proportion of private-sector employment would continue.

## Validation and consensus building

The technical working group, with guidance from WHO experts, held a five-day working session to review, validate, and discuss the data and findings in terms of accuracy and appropriateness. Thereafter, recommendations and concrete policy actions were finalised and submitted to MOH with the view of convening a multisectoral, multistakeholder dialogue on health workforce investment.

## Data sources, validation and quality assurance

Data were triangulated from multiple sources to apply the model (as described above) in Kenya. Table 1 shows a summary of the sources from which data was obtained and inputted into the Microsoft Excel-based model [21] – S1 Data.

## Findings

### Projected health workforce supply in Kenya, 2021–2036

Overall, the supply of the health occupations included in the analysis are projected to expand at an annual average of 3.4%. This rate of increase will likely boost the overall supply from the estimated 194,254 at baseline in 2021 to at least 226,434 by 2025, representing almost 17% improvement from 2021 to 2025. If the trends continue, the supply of health workers could reach 263,676 by 2030, which will be a further 12.6% increase compared to the projected supply for 2025. Additionally, about 15.4% increase from the projected 2030 supply is anticipated by 2035 should the supply dynamics remain fairly constant, bringing the overall supply in 2035 to about 304,351 across the public and private sectors.

**Table 1. Data sources for model application.**

| Dimension for model application | Parameter(s) | Data source(s) |
|---|---|---|
| Population size and demographics | • Population size<br>• Gender distribution<br>• Age composition (age cohorts)<br>• Geographical distribution (county, rural and urban) | • Kenya Population and Housing Census Report and Projections, 2019 |
| Level of health (disease burden) | • Prevalence or incidence of diseases and risk factors that constitute 98% of the burden of mortalities, outpatient attendance and hospital admissions in Kenya<br>• Coverage rates of essential public health interventions. | • Global Burden of Disease Study [31]<br>• Kenya Demographic and Health Survey reports<br>• Midterm review report of Kenya Health Sector Strategic Plan (KHSSP) 2018–2023<br>• Kenya Master Facility List<br>• Kenya Health Information System database<br>• Various peer-reviewed publications |
| Level of service | • The main health services that were being provided or were otherwise necessary to address the diseases and risk factors identified | • Kenya Essential Package of Health Services<br>• Human Resources for Health Norms and Standards 2014<br>• Expert opinion from the technical working group |
| Standard workloads | • The main tasks performed by health workers to address the disease burden identified.<br>• The standard workload per health worker per year is the amount of work within one health service task that one health worker could perform in a year if he/she dedicated all his/her working time to that task. | • Expert opinion from the technical working group<br>• A cross-sectional survey of health professionals [32]. |
| Supply of nurses and midwives | • The existing stock of health professionals, the rate of labour flow (attrition), and the education pipeline (number of admissions into health professions education institutions and pass rates). | • Economic Surveys - 2000–2021 by Kenya National Bureau of Statistics (KNBS)<br>• Human Resources for Health Strategic Plan 2019–2023<br>• National Health Workforce Accounts (NHWA) database<br>• Datasets submitted by professional regulator bodies. |
| Budget and financing data | • Gross domestic product<br>• General Government Health Expenditure (GGHE)<br>• HRH Expenditure as a percent of GGHE<br>• Salaries and income levels of health workers | • Budget Performance Review reports by Controller of Budget<br>• National Health Accounts Report for 2021 |

The nursing and midwifery workforce is, however, projected to expand at a slower pace of 1.5% annually or 7–8% every 5 years if the observed dynamics in attrition (outflows) and inflows remains unchanged. This trajectory will likely lead to a net addition of some 28,000 by 2035. For example, in 2021, there were 109,659 Kenya Registered Community Health Nurses which is projected to increase to about 137,617 by 2035, representing a 25.5% net increase over the period.

Also, there were 18,198 medical laboratory technologists which is projected to increase to 19,697 by 2025 and to 23,023 by 2035. Thus, the medical laboratory technologists are anticipated to increase by 26.5% between 2021 and 2035 if no interventions are made to influence the training capacity and throughputs from the training institutions. Similarly, in 2021 there were 11,129 medical officers, which is projected to expand by just 7% to 11,893 by 2030. If the supply dynamics remain the same, the supply of medical officers is likely to increase by a further 11% to 12,340 in 2035.

Furthermore, the supply of pharmacists is anticipated to increase from 4,069 in 2021 to 5,304 by 2030 and 5,912 by 2035 assuming the training capacity and throughputs remains

similar. The supply of pharmacy technologists, however, is increasing slowly by 0.8% (range: 0.7–1%) annually. There is thus, a seeming de-accelerating in the training outputs of pharmacy technologist leading to an expected 12.2% increase in their supply from 11,429 in 2021 to 12,825 by 2035. Table 2 provides details of the projected supply of 30 categories of health workers in Kenya if the current trend continues without interventions to either abate or accelerate production.

### Projected needs-based requirements for health workers in Kenya to address the population's need for health services

The needs-based projections took into account four main parameters: (a) the disease burden of the country, (b) the size composition of the population along the life course, (c) the package of essential health services required to address the disease burden along the life course of the population and the continuum of public health functions, and (d) the health worker productivity (standard workload). Based on these parameters, the needs-based requirements for health workers in Kenya were projected at 254,220 in 2021 and anticipated to increase at an annual average of 4.7%. If the dynamics of the disease burden and population's demographics remain in the same trajectories, the needs-based requirements could reach 299,452 by 2025 and then 476,278 in 2035.

The projections for doctors, suggest that Kenya required at least 25,100 medical officers in 2021 based on the disease burden, population and service delivery model. However, the need for doctors is projected to triple to 71,643 by 2035. This corresponds to Kenya requiring at least 5 generalist doctors (medical officers) per 10,000 population or approximately one doctor for every 2,000 population to support the aspirations of universal health coverage.

In addition, across medical specialities such as obstetrics and gynaecology, ophthalmology, paediatrics, internal medicine, psychiatry, surgery, and pathology, the country required some 4,863 in 2021. The need for these specialists is further projected to increase by 11.2% by 2025 to 5,427 and to 6,551 by 2030, representing a 17% increase from the 2025 requirements. If the dynamics of the disease burden, composition of the population and the needed health interventions remain the same, the projected need for medical specialist could increase by 23.7% to 8,474 by 2035. These projections corresponds to a ratio of Kenya needing at least one medical specialist per 12,500 population or a density of 8 medical specialists per 100,000 people.

Also, the needs-based requirement for pharmacists and clinical pharmacists was projected to be 5,987 in 2021 across the public and private sectors, which is projected to increase by 16.4% by 2025 to reach 6,970. If the trajectory remains the same, it is projected that the needs-based requirement for pharmacists and clinical pharmacists will expand rapidly by 77% to 12,354 by 2035. The needs-based requirements for the nursing and midwifery workforce were projected to be 142,737 in 2021 and anticipated to increase by 34.3% to 191,639 in 2030. A further 19% increase in the needed nurses and midwives is projected from the 2030 requirement to 228,403 by 2035 across the various disciplines of nursing and midwifery. These projections notwithstanding, if Kenya intends to increase its supply of nurses and midwives to the international labour market, it would be imperative to produce more than the projected needs-based requirements to ensure that outmigration does not compromise the aspiration for universal health coverage. Table 3 provides details of the year-by-year projections of the needs-based requirement for health workers of various.

### Needs versus supply gap analysis

In 2021, the available health workforce in Kenya covered about 76.4% of the needs-based requirements, leaving a gap of 23.6% if the disease burden is to be tackled with appropriate

Table 2. Projected health workforce supply, 2021–2035.

| SN | Health professionals | Projected supply, 2021–2035 | | | | | | | | | | | | | | |
|---|---|---|---|---|---|---|---|---|---|---|---|---|---|---|---|---|
| | | 2021 (Base-line) | 2022 | 2023 | 2024 | 2025 | 2026 | 2027 | 2028 | 2029 | 2030 | 2031 | 2032 | 2033 | 2034 | 2035 |
| 1. | Medical officer | 11,129 | 11,220 | 11,309 | 11,397 | 11,483 | 11,568 | 11,651 | 11,733 | 11,814 | 11,893 | 11,971 | 12,047 | 12,122 | 12,196 | 12,269 |
| 2. | Obstetrician & Gynaecologist | 402 | 454 | 504 | 554 | 603 | 652 | 699 | 746 | 792 | 837 | 881 | 924 | 967 | 1,009 | 1,051 |
| 3. | Ophthalmol-ogist | 104 | 161 | 216 | 271 | 325 | 378 | 430 | 481 | 532 | 581 | 630 | 678 | 725 | 771 | 816 |
| 4. | Paediatrician | 343 | 396 | 447 | 498 | 548 | 598 | 646 | 693 | 740 | 786 | 831 | 876 | 919 | 962 | 1,004 |
| 5. | Physician (Internal Medicine) | 347 | 400 | 451 | 502 | 552 | 601 | 649 | 697 | 744 | 789 | 835 | 879 | 922 | 965 | 1,007 |
| 6. | Psychiatrist | 70 | 127 | 184 | 239 | 293 | 347 | 400 | 451 | 502 | 552 | 601 | 649 | 697 | 744 | 789 |
| 7. | Surgeon | 332 | 397 | 460 | 522 | 584 | 644 | 703 | 761 | 819 | 875 | 930 | 985 | 1,038 | 1,091 | 1,142 |
| 8. | Pathologist | 65 | 122 | 179 | 234 | 289 | 342 | 395 | 447 | 498 | 548 | 597 | 645 | 693 | 740 | 786 |
| 9. | Operating Theatre nurse | N/D | 180 | 357 | 531 | 702 | 870 | 1,035 | 1,198 | 1,357 | 1,514 | 1,668 | 1,820 | 1,969 | 2,116 | 2,260 |
| 10. | Kenya Registered Community Health Nurse | 109,659 | 111,755 | 113,815 | 115,840 | 117,831 | 119,788 | 121,711 | 123,602 | 125,461 | 127,288 | 129,084 | 130,850 | 132,585 | 134,291 | 135,968 |
| 11. | Mental Health/ Psychiatry Nurse | N/D | 72 | 143 | 212 | 281 | 348 | 414 | 479 | 543 | 606 | 667 | 728 | 788 | 846 | 904 |
| 12. | Critical care Nurse | N/D | 122 | 241 | 358 | 474 | 587 | 699 | 808 | 916 | 1,022 | 1,126 | 1,228 | 1,329 | 1,428 | 1,525 |
| 13. | Paediatric Nurse | N/D | 60 | 119 | 177 | 234 | 290 | 345 | 399 | 452 | 505 | 556 | 607 | 656 | 705 | 753 |
| 14. | Kenya Regis-tered Midwife | N/D | 216 | 428 | 637 | 842 | 1,044 | 1,242 | 1,437 | 1,629 | 1,817 | 2,002 | 2,184 | 2,363 | 2,539 | 2,712 |
| 15. | Registered Clinical Officer | 21,797 | 24,216 | 26,595 | 28,933 | 31,231 | 33,490 | 35,711 | 37,893 | 40,039 | 42,149 | 44,222 | 46,260 | 48,264 | 50,233 | 52,169 |
| 16. | Anaesthetist Clinical Officer | 932 | 1,177 | 1,418 | 1,655 | 1,888 | 2,117 | 2,342 | 2,563 | 2,780 | 2,994 | 3,204 | 3,411 | 3,614 | 3,813 | 4,010 |
| 17. | Lung & Skin Clinical Officer | 272 | 357 | 441 | 524 | 605 | 685 | 763 | 840 | 916 | 990 | 1,063 | 1,135 | 1,206 | 1,275 | 1,344 |
| 18. | Paediatric Clinical Officer | 512 | 593 | 673 | 752 | 829 | 905 | 980 | 1,053 | 1,125 | 1,196 | 1,266 | 1,334 | 1,401 | 1,467 | 1,533 |
| 19. | Reproductive Health Clini-cal Officer | 132 | 197 | 261 | 324 | 386 | 447 | 507 | 566 | 624 | 681 | 737 | 792 | 846 | 899 | 951 |
| 20. | Dental surgeon | 1,344 | 1,346 | 1,349 | 1,351 | 1,353 | 1,355 | 1,358 | 1,360 | 1,362 | 1,364 | 1,366 | 1,368 | 1,370 | 1,372 | 1,374 |
| 21. | Community Oral Health Officer | N/D | 41 | 80 | 119 | 158 | 196 | 233 | 269 | 305 | 341 | 375 | 409 | 443 | 476 | 508 |
| 22. | Pharmacist | 4,069 | 4,216 | 4,360 | 4,502 | 4,642 | 4,779 | 4,913 | 5,046 | 5,176 | 5,304 | 5,430 | 5,554 | 5,675 | 5,795 | 5,912 |
| 23. | Pharma-ceutical Technologist | 11,429 | 11,540 | 11,650 | 11,757 | 11,863 | 11,966 | 12,069 | 12,169 | 12,267 | 12,364 | 12,460 | 12,553 | 12,646 | 12,736 | 12,825 |
| 24. | Physiotherapist | 1,757 | 2,114 | 2,465 | 2,810 | 3,149 | 3,483 | 3,810 | 4,132 | 4,449 | 4,760 | 5,066 | 5,367 | 5,663 | 5,953 | 6,239 |
| 25. | Occupational Therapist | 553 | 706 | 856 | 1,003 | 1,148 | 1,290 | 1,431 | 1,568 | 1,704 | 1,837 | 1,967 | 2,096 | 2,222 | 2,347 | 2,469 |
| 26. | Orthopaedic Technologist | 287 | 313 | 338 | 363 | 387 | 411 | 435 | 458 | 481 | 504 | 526 | 547 | 569 | 590 | 610 |

*(Continued)*

**Table 2.** (Continued)

| SN | Health professionals | Projected supply, 2021–2035 | | | | | | | | | | | | | | |
|---|---|---|---|---|---|---|---|---|---|---|---|---|---|---|---|---|
| | | 2021 (Base-line) | 2022 | 2023 | 2024 | 2025 | 2026 | 2027 | 2028 | 2029 | 2030 | 2031 | 2032 | 2033 | 2034 | 2035 |
| 27. | Clinical Dietician | N/D | 650 | 1,289 | 1,917 | 2,534 | 3,141 | 3,737 | 4,324 | 4,900 | 5,467 | 6,024 | 6,571 | 7,109 | 7,638 | 8,158 |
| 28. | Nutritionist | 10,521 | 10,770 | 11,014 | 11,254 | 11,490 | 11,723 | 11,951 | 12,175 | 12,396 | 12,613 | 12,826 | 13,035 | 13,241 | 13,443 | 13,642 |
| 29. | Speech Therapist | N/D | 8 | 17 | 25 | 32 | 40 | 48 | 55 | 63 | 70 | 77 | 84 | 91 | 98 | 105 |
| 30. | Medical Laboratory Technologist | 18,198 | 18,582 | 18,960 | 19,332 | 19,697 | 20,056 | 20,408 | 20,755 | 21,096 | 21,431 | 21,761 | 22,085 | 22,403 | 22,716 | 23,023 |
| | Kenya | 194,254 | 202,507 | 210,620 | 218,595 | 226,434 | 234,140 | 241,715 | 249,161 | 256,481 | 263,676 | 270,749 | 277,702 | 284,536 | 291,255 | 297,859 |
| | % net increase | | 4.2% | 4.0% | 3.8% | 3.6% | 3.4% | 3.2% | 3.1% | 2.9% | 2.8% | 2.7% | 2.6% | 2.5% | 2.4% | 2.3% |

N/D = There was no disaggregated baseline stock data, but annual intake and/or graduate data was available.

interventions from health promotion, disease prevention, detection, treatment, rehabilitation and palliation. The estimated gap translated into a needs-based shortage of 59,966 health workers in 2021. Without appropriate mechanisms to increase the throughput from the education pipeline, increase absorption and retention of the trained health workers, the projections show that the projected supply to the need (the Need Availability Ratio, NAR) could decrease marginally to 75.3% by 2026 and a further decrease to 60.2% by 2035. NAR is a metric of relative gap analysis that compares the supply of the health workforce to the needs-based requirement, indicating the extent to which available health workers meet the health-care needs of a population.

In terms of headcount, Kenya's needs-based shortage of health workers is projected to increase from the estimated 59,966 in 2021 to 114,352 by 2030 and up to 201,581 by 2035 if effective interventions are not implemented to optimise capacity to increase training outputs, strengthen absorption and retention. The projected increase in the needs-based shortage of health workers is partly attributed to the overall health workforce supply increasing at an annual rate of 3.4% compared to an annual increase of 4.7% in the needs-based health work-force requirements, leaving an annual gap of 1.3% between supply and needs.

Underlying the overall projection are significant variations as 31% of the health occupations analysed (n = 10) had the projected supply falling short of meeting even 50% of the needs-based requirement by 2026. In the longer term, the projected supply of about 28% of the health occupations analysed (n = 9) are unlikely to be able to cover half of the needs-based requirements by 2031 unless there are corrective intervention(s) to improve the through-put from the education pipeline. For example, in 2021, there were only 332 surgeons across various surgical sub-specialties which corresponded to just 13.4% of the required 2,475. Even though the projection shows a slightly optimistic trajectory, the anticipated supply of surgeons could cover only 21.4% of the projected needs-based requirements in 2026 and 24.1% in 2030. Under this trajectory, it is expected that the shortage of surgeons will be around 2,931 in 2030. The projections show similar patterns for majority of medical specialists and other specialised health professionals.

On the other hand, 22% of the health occupations analysed (n = 7) seem to have their base-line supply levels in 2021 commensurate with or even surpassing their respective needs-based requirements. For example, the 2021 supply of specialist obstetricians and gynaecologists

**Table 3. Projected needs-based requirements for health workers, 2021–2035.**

| No. | Health professionals | Projected needs-based health workforce requirements | | | | | | | | | | | | | | |
|---|---|---|---|---|---|---|---|---|---|---|---|---|---|---|---|---|
| | | 2021 | 2022 | 2023 | 2024 | 2025 | 2026 | 2027 | 2028 | 2029 | 2030 | 2031 | 2032 | 2033 | 2034 | 2035 |
| 1 | Medical officer | 25,100 | 26,905 | 28,908 | 31,136 | 33,532 | 35,864 | 38,188 | 40,797 | 43,731 | 47,037 | 51,087 | 55,310 | 60,089 | 65,502 | 71,643 |
| 2 | Obstetrician & Gynaecologist | 535 | 547 | 559 | 571 | 583 | 596 | 609 | 623 | 636 | 650 | 665 | 679 | 694 | 709 | 725 |
| 3 | Ophthalmologist | 467 | 487 | 509 | 533 | 560 | 589 | 621 | 657 | 697 | 741 | 791 | 846 | 909 | 979 | 1,059 |
| 4 | Paediatrician | 569 | 549 | 532 | 517 | 504 | 472 | 463 | 456 | 450 | 445 | 426 | 424 | 422 | 422 | 422 |
| 5 | Physician (Internal Medicine) | 426 | 440 | 455 | 472 | 491 | 516 | 539 | 564 | 591 | 622 | 661 | 699 | 742 | 790 | 843 |
| 6 | Psychiatrist | 159 | 156 | 153 | 152 | 150 | 152 | 152 | 152 | 153 | 154 | 157 | 159 | 161 | 163 | 165 |
| 7 | Surgeon | 2,475 | 2,565 | 2,664 | 2,770 | 2,886 | 3,012 | 3,151 | 3,303 | 3,471 | 3,656 | 3,861 | 4,090 | 4,344 | 4,628 | 4,946 |
| 8 | Pathologist | 232 | 237 | 243 | 248 | 253 | 260 | 266 | 271 | 277 | 283 | 291 | 297 | 304 | 310 | 317 |
| 9 | Operating Theatre nurse | 3,560 | 3,639 | 3,719 | 3,801 | 3,884 | 3,969 | 4,057 | 4,146 | 4,237 | 4,330 | 4,426 | 4,523 | 4,622 | 4,724 | 4,828 |
| 10 | Kenya Registered Community Health Nurse | 136,321 | 141,503 | 147,035 | 152,948 | 158,995 | 163,722 | 168,386 | 173,336 | 178,605 | 184,230 | 191,005 | 197,493 | 204,483 | 212,039 | 220,233 |
| 11 | Mental Health/Psychiatry Nurse | 729 | 711 | 697 | 686 | 679 | 680 | 677 | 677 | 678 | 681 | 695 | 701 | 708 | 716 | 726 |
| 12 | Critical care Nurse | 720 | 736 | 752 | 769 | 786 | 803 | 821 | 839 | 857 | 876 | 896 | 915 | 936 | 956 | 977 |
| 13 | Paediatric Nurse | 981 | 979 | 978 | 979 | 982 | 975 | 981 | 989 | 997 | 1,007 | 1,010 | 1,022 | 1,035 | 1,049 | 1,064 |
| 14 | Kenya Registered Midwife | 424 | 433 | 443 | 452 | 462 | 472 | 483 | 493 | 504 | 515 | 527 | 538 | 550 | 562 | 575 |
| 15 | Registered Clinical Officer | 35,101 | 36,120 | 37,245 | 38,487 | 39,857 | 41,604 | 43,279 | 45,131 | 47,182 | 49,456 | 52,236 | 55,056 | 58,202 | 61,718 | 65,655 |
| 16 | Anaesthetist Clinical Officer | 2,992 | 3,058 | 3,126 | 3,195 | 3,265 | 3,337 | 3,411 | 3,487 | 3,564 | 3,643 | 3,724 | 3,806 | 3,891 | 3,977 | 4,066 |
| 17 | Lung & Skin Clinical Officer | 43 | 44 | 45 | 46 | 47 | 49 | 50 | 51 | 52 | 53 | 55 | 56 | 57 | 58 | 60 |
| 18 | Paediatric Clinical Officer | 457 | 457 | 457 | 458 | 459 | 454 | 456 | 459 | 463 | 466 | 465 | 469 | 474 | 480 | 485 |
| 19 | Reproductive Health Clinical Officer | 72 | 74 | 75 | 77 | 79 | 80 | 82 | 84 | 86 | 88 | 90 | 92 | 94 | 96 | 98 |
| 20 | Dental surgeon | 4,145 | 4,237 | 4,330 | 4,425 | 4,522 | 4,687 | 4,790 | 4,896 | 5,004 | 5,114 | 5,310 | 5,427 | 5,546 | 5,668 | 5,793 |
| 21 | Community Oral Health Officer | 1,445 | 1,477 | 1,509 | 1,543 | 1,577 | 1,600 | 1,635 | 1,671 | 1,708 | 1,746 | 1,799 | 1,839 | 1,879 | 1,920 | 1,963 |
| 22 | Pharmacist | 5,094 | 5,273 | 5,468 | 5,682 | 5,919 | 6,211 | 6,502 | 6,824 | 7,183 | 7,584 | 8,063 | 8,566 | 9,130 | 9,764 | 10,478 |
| 23 | Clinical pharmacist | 893 | 927 | 965 | 1,006 | 1,051 | 1,103 | 1,157 | 1,217 | 1,283 | 1,356 | 1,441 | 1,532 | 1,634 | 1,748 | 1,876 |
| 24 | Pharmaceutical Technologist | 4,685 | 4,992 | 5,336 | 5,723 | 6,159 | 6,702 | 7,260 | 7,892 | 8,608 | 9,421 | 10,397 | 11,448 | 12,643 | 14,005 | 15,557 |
| 25 | Physiotherapist | 3,742 | 3,877 | 4,024 | 4,183 | 4,355 | 4,400 | 4,603 | 4,826 | 5,072 | 5,343 | 5,513 | 5,844 | 6,214 | 6,628 | 7,092 |
| 26 | Occupational Therapist | 3,543 | 3,621 | 3,700 | 3,782 | 3,865 | 3,763 | 3,846 | 3,930 | 4,017 | 4,105 | 4,012 | 4,100 | 4,190 | 4,283 | 4,377 |
| 27 | Orthopaedic Technologist | 81 | 82 | 84 | 86 | 88 | 90 | 92 | 94 | 96 | 98 | 100 | 102 | 105 | 107 | 109 |

*(Continued)*

**Table 3.** (Continued)

| No. | Health professionals | Projected needs-based health workforce requirements | | | | | | | | | | | | | | |
|-----|----------------------|------|------|------|------|------|------|------|------|------|------|------|------|------|------|------|
| | | 2021 | 2022 | 2023 | 2024 | 2025 | 2026 | 2027 | 2028 | 2029 | 2030 | 2031 | 2032 | 2033 | 2034 | 2035 |
| 28 | Clinical Dietician | 1,244 | 1,238 | 1,233 | 1,229 | 1,225 | 1,214 | 1,211 | 1,209 | 1,207 | 1,205 | 1,199 | 1,198 | 1,198 | 1,197 | 1,197 |
| 29 | Nutritionist | 5,471 | 5,626 | 5,805 | 6,008 | 6,240 | 6,455 | 6,756 | 7,096 | 7,480 | 7,916 | 8,374 | 8,934 | 9,568 | 10,287 | 11,103 |
| 30 | Speech Therapist | 242 | 248 | 253 | 259 | 264 | 270 | 276 | 282 | 288 | 295 | 301 | 308 | 315 | 322 | 329 |
| 31 | Medical Laboratory Technologist | 11,909 | 12,622 | 13,422 | 14,323 | 15,338 | 16,553 | 17,849 | 19,316 | 20,978 | 22,863 | 25,077 | 27,512 | 30,282 | 33,435 | 37,028 |
| 32 | Orthopaedic Trauma Technologist | 361 | 369 | 377 | 386 | 394 | 403 | 412 | 421 | 430 | 439 | 449 | 459 | 469 | 479 | 490 |
| | **Kenya** | 254,220 | 264,230 | 275,103 | 286,932 | 299,452 | 311,060 | 323,060 | 336,187 | 350,584 | 366,419 | 385,101 | 404,445 | 425,890 | 449,724 | 476,278 |
| | Net increase per year | | 10,009 | 10,873 | 11,828 | 12,521 | 11,607 | 12,001 | 13,126 | 14,397 | 15,835 | 18,682 | 19,344 | 21,445 | 23,834 | 26,554 |
| | % net increase | | 3.9% | 4.1% | 4.3% | 4.4% | 3.9% | 3.9% | 4.1% | 4.3% | 4.5% | 5.1% | 5.0% | 5.3% | 5.6% | 5.9% |
| | Aggregate % change from the baseline | 3.9% | 8.2% | 12.9% | 17.8% | 22.4% | 27.1% | 32.2% | 37.9% | 44.1% | 51.5% | 59.1% | 67.5% | 76.9% | 87.3% | |

matched about 75% of Kenya's needs-based requirement and if the trajectory of training continues and the graduates are retained in the health system, the needs-based shortfall could be offset by 2026. It is, however, important to put in context that a seeming role substitution by reproductive health clinical officers have contributed to mitigate the magnitude of needs-based shortage of specialist obstetricians and gynaecologists.

In 2021, the supply of Kenya Registered Community Health Nurses corresponded to about 80% of the population health needs-based requirements. Nonetheless, the need for nurses is projected to be growing at a rate of about 3.5% percent each year (varying from 3.8–4%) which surpasses the projected 1.5% annual net annual growth in the supply of nurses (varying from 1.2% to 1.9%). This trajectory could lead to an increasing shortage of nurses from 26,662 in 2021 to 61,921 by 2031 if the prevailing capacity of 5,500 is not increased in line with the projected needs-based requirements. The projection further suggest that in 2031 Kenya might have only about 68% of the needs-based requirement for the nursing workforce as the needs continue to outpace the supply.

At baseline in 2021, an estimated 80% of the needs-based requirements for pharmacists was met by the existing supply (5,094 needed vs 4,069 stock) as a result of expansions in the enrolment and throughputs from the pharmacy programmes in various universities. However, as the need increases, the projected supply of pharmacists might match about 77% of the needs-based requirement in 2026 (6,211 vs 4779) and only 67% in 2031 in which the country would potentially face shortfall of 2,634 pharmacists excluding those that specialist pharmacists. Table 4 compares the need versus supply of various health occupations in absolute (shortages or surplus) and relative terms.

## Health workforce financing and economic feasibility analysis of the labour market

In 2021, the public sector of national and county governments potentially allocated US$1.4 billion for health workforce employment (overall wage bill). This budget space is anticipated to expand marginally to US$1.7 billion by 2025. Without additional prioritization of health spending and workforce, economic improvements alone are expected to increase the public

**Table 4. Projected needs-based requirements versus projected supply gap analysis for health workers.**

| No. | Health professionals | 2021 | | | | 2026 | | | | 2031 | | | |
|---|---|---|---|---|---|---|---|---|---|---|---|---|---|
| | | Need (a) | Supply (b) | Gap (b-a) | NAR (b/a) | Need (a) | Supply (b) | Gap (b-a) | NAR (b/a) | Need (a) | Supply (b) | Gap (b-a) | NAR (b/a) |
| 1 | Medical officer | 25,100 | 11,129 | (13,971) | 44.3% | 35,864 | 11,568 | (24,296) | 32.3% | 51,087 | 11,971 | (39,117) | 23.4% |
| 2 | Obstetrician & Gynecologist | 535 | 402 | (133) | 75.2% | 596 | 652 | 56 | 109.3% | 665 | 881 | 216 | 132.6% |
| 3 | Ophthalmologist | 467 | 104 | (363) | 22.2% | 589 | 378 | (211) | 64.2% | 791 | 630 | (161) | 79.7% |
| 4 | Paediatrician | 569 | 343 | (226) | 60.3% | 472 | 598 | 125 | 126.5% | 426 | 831 | 405 | 195.2% |
| 5 | Physician (Internal Medicine) | 426 | 347 | (79) | 81.5% | 516 | 601 | 85 | 116.4% | 661 | 835 | 173 | 126.2% |
| 6 | Psychiatrist | 159 | 70 | (89) | 44.1% | 152 | 347 | 195 | 228.2% | 157 | 601 | 444 | 382.1% |
| 7 | Surgeon | 2,475 | 332 | (2,143) | 13.4% | 3,012 | 644 | (2,368) | 21.4% | 3,861 | 930 | (2,931) | 24.1% |
| 8 | Pathologist | 232 | 65 | (167) | 28.0% | 260 | 342 | 83 | 131.8% | 291 | 597 | 306 | 205.4% |
| 9 | Operating Theatre nurse | 3,560 | – | (3,560) | 0.0% | 3,969 | 870 | (3,099) | 21.9% | 4,426 | 1,668 | (2,757) | 37.7% |
| 10 | Kenya Registered Community Health Nurse | 136,321 | 109,659 | (26,662) | 80.4% | 163,722 | 119,788 | (43,935) | 73.2% | 191,005 | 129,084 | (61,921) | 67.6% |
| 11 | Mental Health/Psychiatry Nurse | 729 | – | (729) | 0.0% | 680 | 348 | (332) | 51.1% | 695 | 667 | (28) | 96.0% |
| 12 | Critical care Nurse | 720 | – | (720) | 0.0% | 803 | 587 | (216) | 73.1% | 896 | 1,126 | 231 | 125.7% |
| 13 | Paediatric Nurse | 981 | – | (981) | 0.0% | 975 | 290 | (685) | 29.7% | 1,010 | 556 | (454) | 55.1% |
| 14 | Kenya Registered Midwife | 424 | – | (424) | 0.0% | 472 | 1,044 | 572 | 221.0% | 527 | 2,002 | 1,475 | 380.1% |
| 15 | Registered Clinical Officer | 35,101 | 21,797 | (13,304) | 62.1% | 41,604 | 33,490 | (8,114) | 80.5% | 52,236 | 44,222 | (8,014) | 84.7% |
| 16 | Anaesthetist Clinical Officer | 2,992 | 932 | (2,060) | 31.1% | 3,337 | 2,117 | (1,221) | 63.4% | 3,724 | 3,204 | (519) | 86.1% |
| 17 | Lung & Skin Clinical Officer | 43 | 272 | 229 | 626.1% | 49 | 685 | 636 | 1405.3% | 55 | 1,063 | 1,009 | 1947.1% |
| 18 | Paediatric Clinical Officer | 457 | 512 | 55 | 112.1% | 454 | 905 | 451 | 199.5% | 465 | 1,266 | 801 | 272.4% |
| 19 | Reproductive Health Clinical Officer | 72 | 132 | 60 | 183.3% | 80 | 447 | 367 | 557.0% | 90 | 737 | 647 | 822.5% |
| 20 | Dental surgeon | 4,145 | 1,344 | (2,801) | 32.4% | 4,687 | 1,355 | (3,332) | 28.9% | 5,310 | 1,366 | (3,944) | 25.7% |
| 21 | Community Oral Health Officer | 1,445 | – | (1,445) | 0.0% | 1,600 | 196 | (1,404) | 12.2% | 1,799 | 375 | (1,424) | 20.9% |
| 22 | Pharmacist | 5,094 | 4,069 | (1,025) | 79.9% | 6,211 | 4,779 | (1,432) | 76.9% | 8,063 | 5,430 | (2,634) | 67.3% |
| 23 | Clinical pharmacist | 893 | – | (893) | 0.0% | 1,103 | – | (1,103) | 0.0% | 1,441 | – | (1,441) | 0.0% |
| 24 | Pharmaceutical Technologist | 4,685 | 11,429 | 6,744 | 243.9% | 6,702 | 11,966 | 5,264 | 178.6% | 10,397 | 12,460 | 2,063 | 119.8% |
| 25 | Physiotherapist | 3,742 | 1,757 | (1,985) | 47.0% | 4,400 | 3,483 | (918) | 79.1% | 5,513 | 5,066 | (446) | 91.9% |
| 26 | Occupational Therapist | 3,543 | 553 | (2,990) | 15.6% | 3,763 | 1,290 | (2,473) | 34.3% | 4,012 | 1,967 | (2,044) | 49.0% |
| 27 | Orthopaedic Technologist | 81 | 287 | 206 | 356.2% | 90 | 411 | 322 | 458.0% | 100 | 526 | 425 | 524.8% |
| 28 | Clinical Dietician | 1,244 | – | (1,244) | 0.0% | 1,214 | 3,141 | 1,927 | 258.7% | 1,199 | 6,024 | 4,825 | 502.5% |
| 29 | Nutritionist | 5,471 | 10,521 | 5,050 | 192.3% | 6,455 | 11,723 | 5,267 | 181.6% | 8,374 | 12,826 | 4,451 | 153.2% |
| 30 | Speech Therapist | 242 | – | (242) | 0.0% | 270 | 40 | (230) | 14.9% | 301 | 77 | (224) | 25.6% |
| 31 | Medical Laboratory Technologist | 11,909 | 18,198 | 6,289 | 152.8% | 16,553 | 20,056 | 3,503 | 121.2% | 25,077 | 21,761 | (3,317) | 86.8% |
| 32 | Orthopedic Trauma Technologist | 361 | – | (361) | 0.0% | 403 | – | (403) | 0.0% | 449 | – | (449) | 0.0% |
| | **Kenya** | **254,220** | **194,254** | **(59,966)** | **76.4%** | **311,060** | **234,140** | **(76,920)** | **75.3%** | **385,101** | **270,749** | **(114,352)** | **70.3%** |

NAR = Need Availability Ratio – a ratio of supply to need, which measures the degree to which the supply of HWF covers the need.

sector budget to $2.2 billion by 2030. In addition, the private sector, a key driver of health employment in Kenya, contributed approximately US$877 million in 2021. This amount could grow to $1.06 billion by 2025 and $1.37 billion by 2030 if the country's economic growth potential is achieved.

Cumulatively, in 2021, the combined financial capacity for the health workforce from both public and private sectors was estimated at $2.29 billion. The financial capacities anticipated to increase by 21% to US$2.77 billion in 2025. If the macroeconomic outlook remains favourable, the financial space could increase by 29.2%, reaching $3.58 billion by 2030.(not worse than 2021). The policy conditions prioritises health in public spending within which the health workforce receives at least the same proportional share of the current health expenditure as before.

In 2021, the wage bill for employing all stock of health workers (at government salary levels) was approximately $2.85 billion on the supply side. Given an unmitigated supply pipeline based on the prevailing trends, the cost of absorbing all those employed from the training pipeline and retaining existing health workers is expected to reach US$3.34 billion in 2025 and up to US$3.9 billion by 2030. To employ all trained health workers in both public and private sectors, a 24.6% increase in financial capacity is needed, or an amount equivalent to 55% of the public sector wage bill. If not addressed, this financial shortfall for health workforce employment may lead to unemployment among skilled health workers, estimated to be 14% in 2021. The 2021 wage bill was approximately 2.62% of GDP (across public and private sectors), but it would have been 3.26% of GDP in 2021 if all unemployed health workers were to be absorbed. Thus, an additional 0.71% of GDP investment is needed to absorb the unemployed health workforce and those in the health professions education pipeline.

Assuming that training would be expanded for the supply of health workers (of various cadres) to meet all the projected needs, the needs-based requirements would have cost some US$4.17 billion in 2021 regarding employment and maintenance of the existing wage bill. Filling the needs-based requirements for health workers represents an average of 4.69% of GDP (range: 4.58–4.83%) between 2021 and 2030 regarding new employment and maintenance of the wage bill across the public and private sectors. Additionally, the cost of training to fill the needs-based gaps (which is shared between the Government and individuals) is about US$761.13 million (ranging from US$510.6 million to US$1.02 billion) or 0.4% of GDP (range: 0.03% to 0.64%). Thus, the health workforce requires an additional investment equivalent to 2.3% of GDP. Table 5 provides summary cost estimates and comparisons, which are graphically illustrated in Fig 2.

## Discussion

This study is the first to apply a needs-based epidemiological approach to health workforce modelling at the national scale covering many health occupations in Kenya. Previous needs-based modelling in Kenya focused on hypertension in rural areas [33]. There are some multi-county estimates, including Kenya, but using different methodologies [2,9,34]. The direction of the findings of this study is consistent with those of previous studies showing needs-based health workforce shortfalls in Kenya. However, our estimation is lower than the World Bank's estimate, which used a global dataset instead of Kenya-specific epidemiological data [34]. Using similar needs-based models but fitted using regional averages [2] also yielded estimates higher than the present study. Thus, the estimates using global averages produce results that are higher than those using regional averages, which are also higher than those using country-specific data.

Comparing the needs-based requirements and the anticipated supply of health workers suggested an absolute shortage of almost 60,000 in 2021, increasing to 114,352 by 2030 and

**Table 5.** Estimates of economic feasibility of supply and needs compared with potential financial space.

| Cost implications and financial sustainability estimates | 2021 | 2025 | 2030 | Average | Minimum | Maximum |
|---|---|---|---|---|---|---|
| Public Sector Budget Space for HWF (a) | 1 413.43 | 1 709.82 | 2 208.31 | 2 104.44 | 1 413.43 | 3 001.88 |
| Estimated Private Sector Contribution (b) | 877.01 | 1 060.92 | 1 370.23 | 1 305.78 | 877.01 | 1 862.63 |
| Cumulative Financial Space (c = a + b) | 2 290.44 | 2 770.74 | 3 578.54 | 3 410.22 | 2 290.44 | 4 864.50 |
| Cost of employing projected supply (d) | 2 853.00 | 3 339.53 | 3 902.61 | 3 718.55 | 2 853.00 | 4 517.57 |
| Cost of employing to fill population health needs-based requirements (f) | 4 173.57 | 4 991.89 | 6 255.87 | 6 101.19 | 4 173.57 | 8 975.88 |
| Cost of training to fill population health needs-based gaps (g) | 510.60 | 644.21 | 811.23 | 761.13 | 510.60 | 1 011.66 |
| Overall investment required based on population health needs (Needs-based Employment + Cost of Training)(f + g) | 4 684.17 | 5 636.11 | 7 067.10 | 6 862.32 | 4 684.17 | 9 987.54 |
| Proportion of supply-side wage bill that could be absorbed by the estimated financial space (d/c) | 80.28% | 82.97% | 91.70% | 90.5% | 80.3% | 107.7% |
| The proportion of population health needs that could be covered by financial space (f/c) | 54.88% | 55.50% | 57.20% | 55.9% | 54.2% | 57.2% |
| Percent of financial space required to absorb "unemployed" health workers | 24.56% | 20.53% | 9.06% | 15.13% | 1.10% | 24.56% |
| Percent of public health sector wage required to absorb "unemployed" health workers | 39.80% | 33.27% | 14.67% | 24.51% | 1.78% | 39.80% |
| Current HWF expenditure as % of GDP | 2.62% | 2.62% | 2.62% | 2.62% | 2.62% | 2.62% |
| Cost of supply as % of GDP | 3.26% | 3.16% | 2.86% | 2.92% | 2.43% | 3.26% |
| Cost of population health needs as % of GDP | 4.77% | 4.72% | 4.58% | 4.69% | 4.58% | 4.83% |
| Additional cost of needs as % of GDP | 2.15% | 2.10% | 1.96% | 2.07% | 1.96% | 2.21% |
| Additional cost of supply as % of GDP | 0.64% | 0.54% | 0.24% | 0.40% | 0.03% | 0.64% |

up to 201,581 by 2035 if nothing is done to boost the system capacity for increased training, absorption and retention. This potential widening in the needs-based shortage of health workers is attributable to a faster rate of expansion in the need for health services due to increasing population and the evolving triple burden of disease on the one hand, and the other hand, a seeming decreasing rate of growth in the supply of health workers. To illustrate this gap, while the aggregate health workforce stock is increasing at a rate of 3.4% (2.7%–4.2%) annually, that of the population's health needs is expanding at 4.7% (3.9%–6.2%) annually.

It is also important to note that the estimated supply of health workforce was based on the observed attrition trend in Kenya. However, with the Government's bilateral agreement to export some 30,000 nurses to the UK and at least 4% attrition of skilled health workers [35], these developments could adversely impact the future supply of health workers. Thus, the imbalances estimated in this paper could be worse if the attrition levels assume a scenario worse than what is modelled. Nevertheless, anticipated expansion in training health workers beyond the current capacity could offset the impact of increased attrition.

The quality of health professions education remains a focal issue among key stakeholders, which can impact the health labour market dynamics. For instance, in Kenya and Uganda, over 33% of surveyed stakeholders articulated reservations concerning the competency levels of recently graduated nurses, opining that they are inadequately equipped to render high-quality patient care [36]. Such data accentuates the imperativeness of integrating a competency-based framework within educational and training curricula, which can improve productivity and boost the supply of health workers [23,37]. To address these concerns and enhance alignment between education and the needs of the health system, Kenya held a conference in 2021 focused on improving and harmonizing health professions education curricula. The conference resulted in actionable recommendations, including strengthening

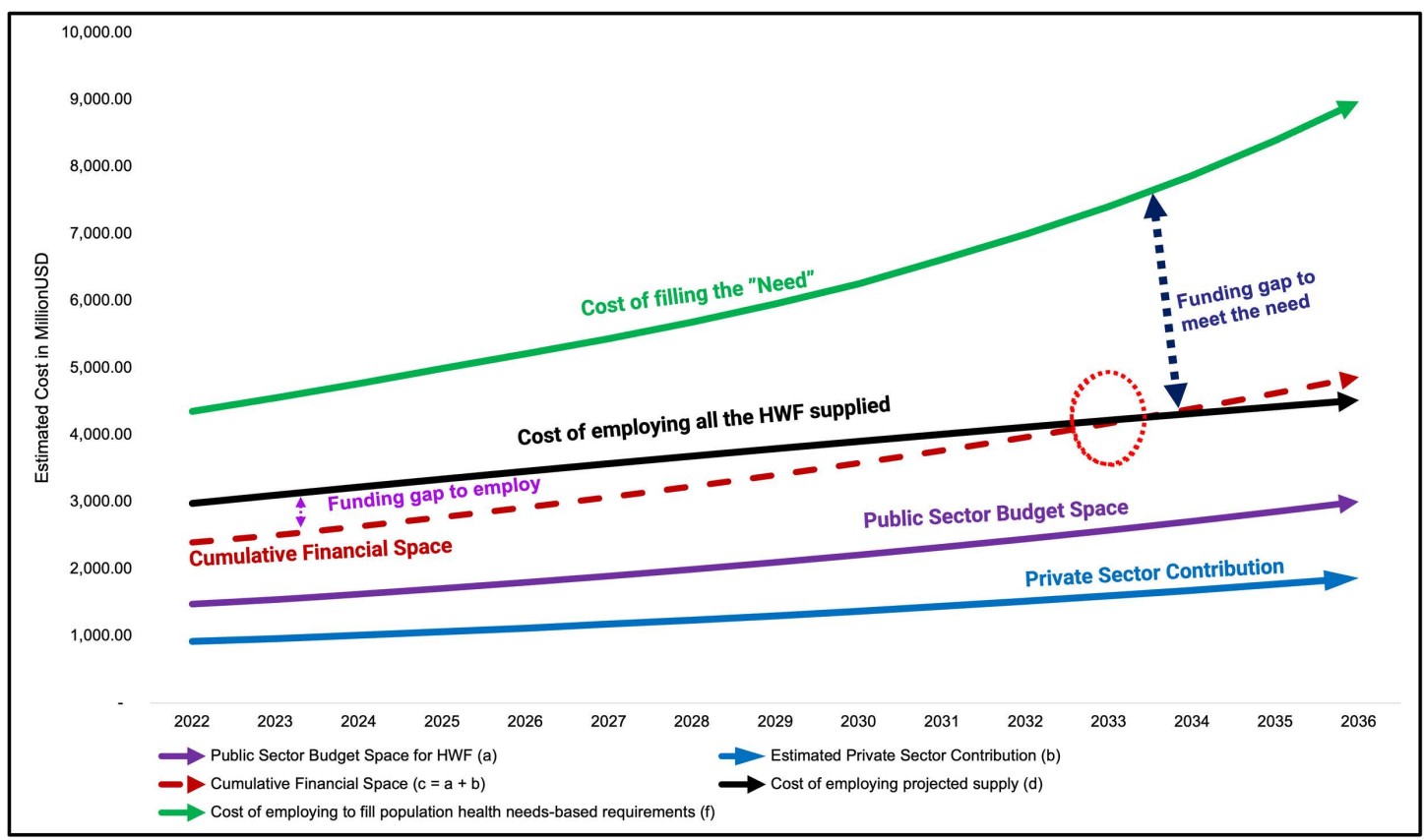

**Fig 2. Financial feasibility analysis under different projection scenarios.**

regulation and accreditation mechanisms, reducing reliance on short-term training programs, and implementing competency-based curricula.

On the levels of investment required in the health workforce, the study highlighted that addressing the labour market mismatches comes with an imperative need for the Kenyan Government, in collaboration with the private sector, to stimulate additional investment in health workforce development and employment by at least 6.5% increase per annum from both the public and private sectors. The private sector's contribution to health workforce employment remains low even when the sector is growing. This increased investment is critical as Kenya is currently grappling with a paradox of unemployment of 14% of the existing health workforce [14] amidst a conspicuous deficit in health workers in the frontline of service delivery. A demographic surge in Kenya is escalating the demand for health services, necessitating an expansion in the health workforce. While the study anticipates that the private sector will absorb a proportionate share of the newly trained healthcare workers, the role of the public sector remains pivotal.

The study found that as of 2021, the existing health workers cover about 76.4% of the Kenyan population's health needs (considering country-specific disease burden, demographic profile, essential health services and professional standards for service delivery), but the HWF unemployment rate of at least 14%. If the production of health workers and budgetary prioritisation of HWF remains constant over time, the budget space analysis suggests that there could be an annual financing gap of 15.13% that could result in unemployment. However, Kenya is not an isolated case. For example, comparable data from ten African countries

derived through HLMAs suggest that almost 27% of the trained health workers might be unemployed or underemployed [3].

The paradox of having high unemployment among health workers despite a needs-based shortage, is a product of a misalignment between workforce supply and ability and willingness to employ (demand) [38,39]. This paradox often arises from low financial space due to restrictive public financial management rules that puts ceilings on public sector employment while stakeholders are expanding the training with the view to meeting planning targets that are delinked from the budget ceilings [8,40,41]. This underscores need for a paradigm shift from a one-size-fits-all notion that training more is also "the main solution" to needs-based shortage of health workers. Expanding budget space allocation by the ministry of finance and/or the approved establishment by the custodians of public employment are equally critical.

Undoubtedly, Kenya needs to urgently develop a master training plan to optimise and align the training pipeline with the needs of the population (as projected). This may require scaling up training in specific occupations while maintaining the current training capacity for others. However, there is greater attention on achieving a wage bill ceiling target of 35% and decision-makers are exploring ways of cutting down the public wage bill (the numerator) [17]. With the enormous return on investment of US$9 for every US$1 spent [42], it could be progressive to explore expanding the health budget (the denominator) through innovative financing mechanisms and efficiency gains from the almost 1 in 4 dollars of the health spending lost to technical inefficiency in the health system [43]. In addition, Kenya could explore using the principles of the Africa health workforce investment charter [44] to align and stimulate health workforce investments with the view of expanding employment by the devolved county governments and the private sector through a mutually agreed investment compact – as these entities make independent decisions on the employment of health workers. In addition, a national staffing norm/standard could be strategically used to provide both financial and regulatory incentives to drive job creation. For example, financial incentive could be provided through enhanced reimbursement rates by the health insurance scheme for service performed by those with adequate staffing. Regulatory enforcement for health facilities to meet the set staffing standards, particularly for the private sector could also drive job creation [45].

Furthermore, at the heart of addressing the broader health workforce challenges is the management of the health workforce that can contribute to improving efficiency [46] and health sector industrial harmony [47,48]. There is the need to attract and retain health workers, particularly in rural and underserved areas where health services are often less than optimal. To achieve this, the MOH needs to collaboratively work with the county governments and other ministries to develop competitive incentives, supportive work environments, and professional development opportunities that can attract and motivate health workers. Such benefits have been shown to improve retention rates and encourage health workers to remain and serve where they are needed most, fostering a stable and committed workforce [49].

Finally, Kenya can leverage partnerships with international health organizations, private sectors, and academic institutions to secure funding, resources, and technical support. These collaborations can bridge financial gaps and provide vital expertise in healthcare training and workforce distribution, accelerating Kenya's progress toward an equitable health system.

## Limitations

One observation worth noting is the need for a clearly distinguished scope of practice across clinical officers, doctors, and, to some extent, nurses/midwives. This made it difficult to accurately determine activities or interventions exclusively carried out by doctors and/or clinical officers at the primary and secondary levels of care. There will likely be some overlap between

the estimated needs-based requirement for doctors and clinical officers. With this caveat, based on the prevailing health service delivery model, the need for clinical officers (of all generalist and specialised ones) appears to be on the ascendency, increasing by 13% from 38,665 in 2021 to 43,707 by 2025 and a further 23% increase to 53,706 by 2030. If the factors affecting the need for clinical officers remain the same, their need could be more than 70,000 by 2035, an increase of 31% from the number needed in 2030. This could be ameliorated if task-shifting and/or accelerating the training of doctors is pursued.

Even though the tool used for the analysis is able to run sensitivity analysis (in the form of best and worst case scenarios) [4], it relies on reported confidence intervals of the prevalence rates of diseases. However, available data at the time of analysis were point estimates without information on the boundaries of uncertainty. As a result, it was not feasible to conduct sensitivity analysis which should be considered a limitation of this paper. Strengthening routine health information system would be critical for enhancing the precision and ability to run robust sensitivity analysis in subsequent updates.

The input data for this modelling study was triangulated from various sources as highlighted in the methodology section. As a result, the underlying limitations and assumptions of these data sources are indirectly inherited by this study. Some of these limitations may include, but not limited to variations in the data collection methods, rigor, frequency of updates, and levels of accuracy, which may lead to inconsistencies.

## Supporting information

**S1 Data. Health workforce needs-based analysis tool.**
(XLSM)

## Acknowledgments

The Ministry of Health (MOH) Health Labour Market Analysis (HLMA) Technical Working Group (TWG) contributed to data collection.

## Author contributions

**Conceptualization:** James Avoka Asamani, Brendan Kwesiga, Evalyne Chagina, Zeinab Gura, Nakato Jumba, Mutile Wanyee, Njoroge Nyoike, Juliet Nabyonga-Orem, Paul Marsden, Mona Almudhwahi Ahmed, Pascal Zurn, Annah Wamae.

**Data curation:** James Avoka Asamani, Brendan Kwesiga, Sunny C. Okoroafor, Evalyne Chagina, Joel Gondi, Zeinab Gura, Francis Motiri, Nakato Jumba, Teresa Ogumbo, Nkatha Mutungi, Stephen Muleshe, Yusuf Suraw, Hanah Gitungo, Kiogora Gatimbu, Mutile Wanyee, Amos Oyoko, Angela Nyakundi, Stephen Kaboro, Mary Wanjiru Njogu, Maureen Monyoncho, Njoroge Nyoike, Wesley Ogera Ooga, Julius Korir, Julius Ogato, Annah Wamae.

**Formal analysis:** James Avoka Asamani, Brendan Kwesiga, Sunny C. Okoroafor, Evalyne Chagina, Joel Gondi, Francis Motiri, Nakato Jumba, Teresa Ogumbo, Stephen Muleshe, Yusuf Suraw, Hanah Gitungo, Kiogora Gatimbu, Mutile Wanyee, Amos Oyoko, Angela Nyakundi, Stephen Kaboro, Mary Wanjiru Njogu, Maureen Monyoncho, Njoroge Nyoike, Wesley Ogera Ooga, Julius Korir, Paul Marsden, Julius Ogato, Pascal Zurn, Annah Wamae.

**Funding acquisition:** James Avoka Asamani, Brendan Kwesiga, Mona Almudhwahi Ahmed, Pascal Zurn.

**Investigation:** James Avoka Asamani, Teresa Ogumbo, Stephen Muleshe, Amos Oyoko, Angela Nyakundi, Mary Wanjiru Njogu, Wesley Ogera Ooga, Julius Korir, Mona Almudhwahi Ahmed, Julius Ogato.

**Methodology:** James Avoka Asamani, Francis Motiri, Amos Oyoko, Njoroge Nyoike, Wesley Ogera Ooga, Juliet Nabyonga-Orem, Julius Korir, Paul Marsden, Pascal Zurn.

**Project administration:** Brendan Kwesiga, Joel Gondi, Zeinab Gura, Mutile Wanyee, Juliet Nabyonga-Orem, Paul Marsden, Mona Almudhwahi Ahmed, Annah Wamae.

**Resources:** Sunny C. Okoroafor.

**Software:** James Avoka Asamani.

**Supervision:** Nakato Jumba, Juliet Nabyonga-Orem, Mona Almudhwahi Ahmed, Julius Ogato, Pascal Zurn, Annah Wamae.

**Validation:** James Avoka Asamani, Brendan Kwesiga, Sunny C. Okoroafor, Evalyne Chagina, Joel Gondi, Francis Motiri, Teresa Ogumbo, Stephen Muleshe, Kiogora Gatimbu, Mutile Wanyee, Stephen Kaboro, Mary Wanjiru Njogu.

**Writing – original draft:** James Avoka Asamani, Julius Korir.

**Writing – review & editing:** James Avoka Asamani, Brendan Kwesiga, Sunny C. Okoroafor, Evalyne Chagina, Joel Gondi, Zeinab Gura, Francis Motiri, Nakato Jumba, Teresa Ogumbo, Nkatha Mutungi, Stephen Muleshe, Yusuf Suraw, Kiogora Gatimbu, Mutile Wanyee, Amos Oyoko, Angela Nyakundi, Stephen Kaboro, Mary Wanjiru Njogu, Maureen Monyoncho, Njoroge Nyoike, Wesley Ogera Ooga, Juliet Nabyonga-Orem, Julius Korir, Paul Marsden, Mona Almudhwahi Ahmed, Julius Ogato, Pascal Zurn, Annah Wamae.

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
