## [Decision Letter · Decision Letter 0]

25 Oct 2024

PGPH-D-24-02288

Modelling the health labour market outlook in Kenya: supply, needs and investment requirements for health workers, 2021 - 2035

Dear Dr. Asamani,

Thank you for submitting your manuscript to PLOS Global Public Health. After careful consideration, we feel that it has merit but does not fully meet PLOS Global Public Health’s publication criteria as it currently stands. Therefore, we invite you to submit a revised version of the manuscript that addresses the points raised during the review process.

Kindly discuss and elaborate on some of your findings as suggested by the reviewers. For instance, discussion should provide more insight and recommendation that would build on the information found in the models and why Kenya actively plans to export nurses when there are needs at home. Such detail will provide important clarity to this work and improve its quality.

We look forward to receiving your revised manuscript.

Kind regards,

Ikechi G Okpechi

Academic Editor

Journal Requirements:

Additional Editor Comments (if provided):

Thank you for submitting your manuscript to PLOS Global Public Health. After careful consideration, we feel that it has not fully meet PLOS Global Public Health’s publication criteria as it currently stands. Therefore, we invite you to submit a revised version of the manuscript that addresses the points raised during the review process.

The manuscript has been evaluated by three reviewers, and their comments are available below.

The reviewers have requested a couple of major revisions. Could you please carefully revise the manuscript to address all comments raised?

Reviewers' comments:

Reviewer's Responses to Questions

**Comments to the Author**

1. Does this manuscript meet PLOS Global Public Health’s publication criteria ? Is the manuscript technically sound, and do the data support the conclusions? The manuscript must describe methodologically and ethically rigorous research with conclusions that are appropriately drawn based on the data presented.

Reviewer #1: Yes

Reviewer #2: Yes

Reviewer #3: Yes

2. Has the statistical analysis been performed appropriately and rigorously?

Reviewer #1: Yes

Reviewer #2: I don't know

Reviewer #3: Yes

3. Have the authors made all data underlying the findings in their manuscript fully available (please refer to the Data Availability Statement at the start of the manuscript PDF file)?

Reviewer #1: Yes

Reviewer #2: Yes

Reviewer #3: Yes

4. Is the manuscript presented in an intelligible fashion and written in standard English?

Reviewer #1: Yes

Reviewer #2: Yes

Reviewer #3: Yes

5. Review Comments to the Author

Reviewer #1: This was a comprehensive modelling study about Kenya’s healthcare workforce. This is an extremely important study for Kenya’s national healthcare planning. They used a published simulation tool and convened a technical expert panel to generate the necessary inputs. The equations underlying the models are well described.

I have just a few comments:

1) It would be helpful to know if the excel file used is a widely used model for HLMA and whether other models were considered.

2) The supplementary excel file is very comprehensive, but it is a bit unclear where the inputs came from for every sheet. The addition of some footnotes would be helpful. In Table 1, are the data sources cites such at the reader can visit each of the respective websites/reports?

3) “Despite progress, the WHO Africa Region needs more than 6.1 million alongside the maldistribution of available health workers, one of the most significant obstacles to achieving UHC by 2030” – consider rewording sentence to improve readability

4) WHO Africa Region vs. WHO African Region – be consistent about Africa vs. African

5) Introduction “absorption capacity” – define this in the introduction when it first used, as it may be unfamiliar to many readers

6) Results, “range from 4.2% in 2021 to 2.7% by 2030” – Explain in the methods where this range comes from. Is this using a range of model inputs (like a sensitivity analysis using 80% to 125% the input values), or inputs from multiple sources? Is this by varying one input at a time, or all inputs simultaneously? Etc.

7) In general, were there sensitivity analyses that were performed within the Excel file model? Describe in methods.

Reviewer #2: Asamani et al. modeled the health labor market outlook in Kenya and found that there is a need for an increased health workforce and an increase in health workforce budget to employ these skilled workers.

Such analyses are important for countries to take stock and understand the projected needs and costs to permit early intervention, as these will only have future consequences.

I am not a modeler and cannot comment on the equations developed and used, although the methodology does seem intuitively plausible. The findings are relatively straight forward. For a general reader however there is some depth lacking that would be helpful to expand. The discussion is very short. There are therefore some clarifications that would be required:

1. Do the costs for the health care workers include the costs of training? If not, why not?

2. It is mentioned several times that a proportion of health care workers (nearly 1/3 mentioned in the introduction, 14% mentioned in the discussion – the numbers should be consistent throughout?) remain unemployed – please expand here, is this simply because of lack of funds to pay salaries or are there other drivers which may need to be considered in the models? Is this across all cadres or only some? If some, then which? This may be helpful to anticipate some redistribution of training resources

3. There is brief mention of concerns about the quality of nursing training – is this a funding issue or why is this? Are people being rushed through to meet a need? This would be counterintuitive if some nurses are remaining unemployed (unless unemployable)

4. Why does a country actively plan to export nurses if there is a need at home?

5. In theory, if the health care workforce is well trained and adequate could one envisage the disease burden declining? How would this impact the modelling?

6. Page 6, 2nd line of 1st paragraph, word missing “in the labour in the future”

7. Please suggest/hypothesise what an optimal approach/solution would be for Kenya to meet its own needs?

8. the Limitations are very focused, there are other potential sources of error in the data sets used, please mention these.

Reviewer #3: Observed grammatical/ typo mistakes

Page 4: - Over the last decade / then over the last decades – Perhaps be more concise for over the last decades – Suggest state – the years

page 5: 1. Grammar: "The descriptive aspects of the health labour market analysis has been discussed..." should be corrected to "have" .

page 13. Grammar

• "at annual average" should be "at an annual average."

• "the estimate 194,254" should be "the estimated 194,254."

• "are anticipated to by 26.5%" should be "are anticipated to increase by 26.5%."

page 16: Grammar

• “requirements for health workers in Kenya was projected at” → should be “were projected at.”

• "If the dynamics of the disease burden and population’s demographics remains..." → should be "populations' demographics remain..."

Page 17; Grammar/typos

• “The needs-based requirements for the nursing and midwifery workforce was projected...” → should be “were projected.”

• “A further 19% increase in the needed nurses and midwives id projected...” → “is projected.”

• “...projected from the the 2030 requirement...” → should be "the 2030 requirement

page 20

Grammar:

o “...if the disease burden burden is to be tackled” → Repetition of “burden.”

o “Without appropriate mechanisms to increase the throughput from the education pipeline, increase absorption and retention of the trained health workers, the projection show...” → should be "the projections show."

Other:

Consider explaining/ defining the Need Availability Ratio (NAR) – as many readers will not be familiar with this

Recommendations - discussion

Discuss why there’s a high unemployment rate despite health workforce shortages and propose specific solutions to reduce it.

I believe that the discussion should provide more insight and recommendation that would build on the information found in the models

For example:

1. More detail clarifying the roles of various stakeholders ( ex governments, private institution etc) in addressing the employment problems

2. Provide solutions or strategies for increasing training output and retention, which could be highlighted further, such as improving incentives for healthcare workers.

3. Discuss the roll that public/private partnerships may provide

4. Provide more detail on proposed strategies to close the 43$ funding gap and better integrate health workers into the work force - especially in rural and underserved areas

4. Page 4: The increase in the wage bill from KSh 4.97 billion to KSh 7.14 billion (2014–2019) More details are needed on why health outcomes have not met expectations despite these investments.- if known

5. Consider mentioning specific strategies to increase retention, such as government policies or funding for education, to provide actionable recommendations.

6 Highlight strategies for mitigating attrition and increasing training to meet health needs.

Potential risks: It would be nice to have insight in what the future concerns are:

1. Example: It may be helpful to highlight any potential risks or concerns if supply trends are not adjusted to meet future demands.

6. PLOS authors have the option to publish the peer review history of their article (what does this mean? ). If published, this will include your full peer review and any attached files.

**Do you want your identity to be public for this peer review?** For information about this choice, including consent withdrawal, please see our Privacy Policy .

Reviewer #1: No

Reviewer #2: No

Reviewer #3: **Yes: ** Nicola Wearne

---

## [Decision Letter · Decision Letter 1]

4 Dec 2024

PGPH-D-24-02288R1

Modelling the health labour market outlook in Kenya: supply, needs and investment requirements for health workers, 2021 - 2035

Dear Dr. Asamani,

Thank you for submitting your manuscript to PLOS Global Public Health. After careful consideration, we feel that it has merit but does not fully meet PLOS Global Public Health’s publication criteria as it currently stands. Therefore, we invite you to submit a revised version of the manuscript that addresses the points raised during the review process. Comments from Reviewer #3 can be seen below.

We look forward to receiving your revised manuscript.

Kind regards,

Ikechi G Okpechi

Academic Editor

Journal Requirements:

Additional Editor Comments (if provided):

Reviewers' comments:

Reviewer's Responses to Questions

**Comments to the Author**

1. If the authors have adequately addressed your comments raised in a previous round of review and you feel that this manuscript is now acceptable for publication, you may indicate that here to bypass the “Comments to the Author” section, enter your conflict of interest statement in the “Confidential to Editor” section, and submit your "Accept" recommendation.

Reviewer #1: All comments have been addressed

Reviewer #2: All comments have been addressed

Reviewer #3: All comments have been addressed

2. Does this manuscript meet PLOS Global Public Health’s publication criteria ? Is the manuscript technically sound, and do the data support the conclusions? The manuscript must describe methodologically and ethically rigorous research with conclusions that are appropriately drawn based on the data presented.

Reviewer #1: Yes

Reviewer #2: Yes

Reviewer #3: Yes

3. Has the statistical analysis been performed appropriately and rigorously?

Reviewer #1: Yes

Reviewer #2: I don't know

Reviewer #3: Yes

4. Have the authors made all data underlying the findings in their manuscript fully available (please refer to the Data Availability Statement at the start of the manuscript PDF file)?

Reviewer #1: Yes

Reviewer #2: Yes

Reviewer #3: Yes

5. Is the manuscript presented in an intelligible fashion and written in standard English?

Reviewer #1: Yes

Reviewer #2: Yes

Reviewer #3: Yes

6. Review Comments to the Author

Reviewer #1: All comments have been thoroughly addressed.

Reviewer #2: My queries have been answered

Reviewer #3: Just a few comments

In the discussion : the following is very cumbersome: suggest rewriting : As part of efforts to address some of these concerns and foster alignment between education and the needs of the health system, in 2021, Kenya held a conference to improve and harmonize health professions education curricula which led to actionable recommendations such as strengthening regulation and accreditation mechanisms, minimizing quick-fix trainings, and use of competency-based curricula.

Example: To address these concerns and enhance alignment between education and the needs of the health system, Kenya held a conference in 2021 focused on improving and harmonizing health professions education curricula. The conference resulted in actionable recommendations, including strengthening regulation and accreditation mechanisms, reducing reliance on short-term training programs, and implementing competency-based curricula.

Mistake noted:

As a result, it was not feasible to conduct sensitivity analysis which should be considered a limitation "if"should be replaced by "of " this paper.

7. PLOS authors have the option to publish the peer review history of their article (what does this mean? ). If published, this will include your full peer review and any attached files.

**Do you want your identity to be public for this peer review?** For information about this choice, including consent withdrawal, please see our Privacy Policy .

Reviewer #1: No

Reviewer #2: No

Reviewer #3: **Yes: ** Nicola Wearne

---

## [Editor Report · Decision Letter 2]

23 Dec 2024

Modelling the health labour market outlook in Kenya: supply, needs and investment requirements for health workers, 2021 - 2035

PGPH-D-24-02288R2

Dear Prof Asamani,

We are pleased to inform you that your manuscript 'Modelling the health labour market outlook in Kenya: supply, needs and investment requirements for health workers, 2021 - 2035' has been provisionally accepted for publication in PLOS Global Public Health.

Best regards,

Ikechi G Okpechi

Academic Editor
